# Immune Microenvironment and the Effect of Vascular Endothelial Growth Factor Inhibition in Hepatocellular Carcinoma

**DOI:** 10.3390/ijms252413590

**Published:** 2024-12-19

**Authors:** Kyoko Oura, Asahiro Morishita, Tomoko Tadokoro, Koji Fujita, Joji Tani, Hideki Kobara

**Affiliations:** Department of Gastroenterology and Neurology, Faculty of Medicine, Kagawa University, 1750-1 Ikenobe, Miki, Kita 761-0793, Kagawa, Japan; oura.kyoko@kagawa-u.ac.jp (K.O.);

**Keywords:** hepatocellular carcinoma, immune microenvironment, vascular endothelial growth factor, atezolizumab, bevacizumab, durvalumab, tremelimumab, immune check-point inhibitor, immune cycle, regulatory T cell

## Abstract

Systemic therapy for unresectable hepatocellular carcinoma (HCC) has progressed with the development of multiple kinases, such as vascular endothelial growth factor (VEGF) signaling, targeting cancer growth and angiogenesis. Additionally, the efficacy of sorafenib, regorafenib, lenvatinib, ramucirumab, and cabozantinib has been demonstrated in various clinical trials, and they are now widely used in clinical practice. Furthermore, the development of effective immune checkpoint inhibitors has progressed in systemic therapy for unresectable HCC, and atezolizumab + bevacizumab (atezo/bev) therapy and durvalumab + tremelimumab therapy are now recommended as first-line treatment. Atezo/bev therapy, which combines an anti-programmed cell death 1 ligand 1 antibody with an anti-VEGF antibody, is the first cancer immunotherapy to demonstrate efficacy against unresectable HCC. With the increasing popularity of these treatments, VEGF inhibition is attracting attention from the perspective of its anti-angiogenic effects and impact on the cancer-immune cycle. In this review, we outline the role of VEGF in the tumor immune microenvironment and cancer immune cycle in HCC and outline the potential immune regulatory mechanisms of VEGF. Furthermore, we consider the potential significance of the dual inhibition of angiogenesis and immune-related molecules by VEGF, and ultimately aim to clarify the latest treatment strategies that maximizes efficacy.

## 1. Introduction

Primary liver cancers are among the major malignant tumors worldwide, ranking sixth in terms of incidence and third in mortality [1]. Among them, hepatocellular carcinoma (HCC), which originates in the hepatocytes, is the most common type, accounting for 90% of cases [2], and is caused by persistent hepatitis virus infection, such as infection with the hepatitis B virus (HBV) and hepatitis C virus (HCV), metabolic dysfunction-associated steatotic liver disease [3,4], and alcoholic liver disease [5,6]. Treatment options are selected based on the stage of HCC, liver function, and the patients’ general condition, including surgical resection, liver transplantation, ablation, and transarterial chemoembolization (TACE) [7]. HCC recurs frequently and eventually progress to the stage where these treatments are no longer applicable. Unresectable HCC with metastasis and/or major vascular invasion is classified as stage C according to the Barcelona Liver Cancer (BCLC) Staging System, and systemic therapy is strongly recommended [7]. Furthermore, patients with BCLC stage B disease who show resistance to conventional TACE are also treated with systemic therapy, increasing opportunities to receive such therapy [8].

Since the efficacy of sorafenib, a molecular targeted agent (MTA), for unresectable HCC was first reported in 2008 [9], systemic therapy for HCC has progressed with the development of MTAs that target multiple kinases related to angiogenesis and cancer proliferation, including the vascular endothelial growth factor (VEGF) signaling pathway. Furthermore, the efficacies of regorafenib [10], lenvatinib [11], ramucirumab [12], and cabozantinib [13] have been demonstrated in phase III trials and have been widely applied in clinical practice. Additionally, the development of effective immune checkpoint inhibitors (ICIs) has progressed as a systemic therapy for unresectable HCC. Atezolizumab plus bevacizumab (atezo/bev) [14] or durvalumab plus tremelimumab therapy [15], which are regimens including ICIs, has been suggested as the first-line treatments for unresectable HCC. Specifically, atezo/bev therapy, which combines an anti-programmed cell death 1 (PD-1) ligand 1 (PD-L1) antibody with a VEGF antibody, was the first immunotherapy to show efficacy against unresectable HCC [16]. As this type of combination therapy becomes more widespread, attention is being paid not only to the significance of VEGF inhibition in terms of its effect on inhibiting angiogenesis, but also to its effect on the cancer immune cycle.

The tumor microenvironment (TME) of HCC is a complex and dynamic system composed of tumor cells, immune cells, endothelial cells, fibroblasts, stromal cells, and extracellular matrix (ECM) components. In this context, VEGF plays a pivotal role not only in promoting angiogenesis but also in modulating immune responses, fostering tumor progression, and inducing therapeutic resistance. First, VEGF promotes angiogenesis and changes the tissue environment. In HCC, the new blood vessels formed by VEGF are often structurally abnormal and highly leaky, which increases interstitial pressure within the tumor, impairs drug delivery, and creates a hypoxic environment [17,18]. This hypoxic state further stimulates the production of VEGF, forming a feedback loop that exacerbates angiogenesis and tumor growth. Second, VEGF promotes tumor immune escape by mobilizing and suppressing immune cells and reducing the function of antigen-presenting cells, thereby forming an immunosuppressive TME [18,19]. Third, VEGF modulates communication between tumor cells and immune cells. The activation of the VEGF signaling pathway leads to the abnormal secretion of cytokines and chemokines, which regulates the mobilization and localization of immune cells [19]. In addition, VEGF induces fibroblast activation and ECM remodeling, accelerating tumor cell invasion and metastasis [20]. Therefore, elucidating the complex mechanisms of the immune microenvironment of HCC, rather than simply inhibiting angiogenesis through VEGF inhibition, may lead to the development of therapeutic targets and the overcoming of treatment resistance.

However, previous studies have provided fragmented insights into the changes in the immune microenvironment of HCC induced by VEGF inhibition, failing to provide a comprehensive understanding of its effects when combined with ICIs. This review aims to systematically synthesize the immunomodulatory effects of VEGF inhibition from the perspective of the TME and comprehensively discuss the role of VEGF in the cancer immunity cycle. Additionally, it critically evaluates existing research findings and highlights their limitations regarding the combination of VEGF inhibition and immunotherapy. This approach is expected to contribute to the identification of novel therapeutic targets and the development of strategies to overcome treatment resistance in HCC.

## 2. Cancer Immunity Cycle

Favorable clinical outcomes have been reported for various cancers treated with antibodies targeting PD-1, PD-L1, and cytotoxic T-lymphocyte-associated protein 4 (CTLA-4), prompting increased interest in cancer immunology [21,22]. The TME, comprising the tumor, ECM, and various immune cell subsets, plays a pivotal role in modulating cancer progression and therapeutic response [23]. Understanding the cancer immunity cycle, a seven-step process essential for immune-mediated cancer control, is critical for comprehending immune function within the TME [24,25]. Table 1 provides on an overview of this cycle.

In Step 1, cancer antigens are released from dead cancer cells through proliferative necrosis of therapeutic intervention. In Step 2, dendritic cells (DCs) capturing these antigens migrate to the lymph nodes (LNs), process them into specific peptides on major histocompatibility complex (MHC) class I molecules, and present them to CD8+ cytotoxic T lymphocytes (CTLs). In Step 3, CTLs undergo priming and activation. In Step 4, activated CTLs migrate to tumor tissue, and in Step 5, they infiltrate the tumor tissue. In Step 6, CTLs recognize cancer cells via T-cell receptors (TCRs) and engage costimulatory receptors, including CD28 and CD137, binding to ligands on DCs. In Step 7, CTLs attack cancer cells, completing the cycle. In HCC, dysfunctions in cancer immunity contribute to immune evasion, tumor progression, and treatment resistance. A major immune escape mechanism involves inhibiting T-cell activation through interactions between PD-1 on T cells and PD-L1 on cancer cells [26,27], as well as between CTLA-4 on T cells and CD80/CD86 molecules on DCs [28], which are key targets of current immunotherapies.

The immunological phenotypes of tumors, crucial for predicting immunotherapy success, are classified into three categories: inflamed, immune-excluded, and immune-desert types [23]. The inflamed type promotes CTL infiltration through immune activation, including tertiary lymphoid structures near within the tumor. In the immune-excluded type, suppressive stroma and ECM prevent CTLs from reaching cancer cells, limiting immune infiltration to the stroma rather than the tumor parenchyma. The immune-desert type is characterized by an absence of immune cells in the TME, reflecting a lack of immune cell recruitment and infiltration [29,30]. These immunotype vary across cancers; for example, melanoma frequently exhibits the inflamed type, colon cancer the immune-excluded type, and prostate cancer the immune-desert type [29]. Since immunotypes can change due to tumor progression or therapeutic intervention [31], understanding factors contributing to immune-excluded or immune-desert phenotypes is essential for predicting therapeutic outcomes and guiding treatment strategies.

## 3. Immune Microenvironment in HCC

### 3.1. Immunostimulative Roles of CTLs

CTLs, T cells that express CD8 on their cell membranes, are key players in the immune cycle. Naïve CD8+ T cells lack cytotoxic activity until their TCRs detect cancer antigens presented by MHC class I molecules on antigen-presenting cells. Upon receiving signals from costimulatory molecules, naïve CD8+ T cells become activated, differentiating into CTLs with cytotoxic functions against antigen-presenting cancer cells [23]. CTLs produce tumor necrosis factor (TNF), granzyme, and perforin to exert cytotoxic activity against HCC cells, and some remain memory-killer cells, carrying a mechanism that recognizes the same cancer cells.

In the TMEs of HCC, the high expression of several regulatory molecules including VEGF, C-X-C motif chemokine ligand 17, indoleamine 2,3-dioxygenase (IDO), and interleukin-10 (IL-10), the lack of CD4+ cells, metabolic competition with HCC cells competition, and hypoxia limit the specific responses to tumor-associated antigens and cause poor release of interferon (IFN)-γ by CTLs, and the presence of CTLs in HCC has positive impact on the prognosis for patients [32,33,34,35]. Furthermore, the expression of Fas and its ligands in CTLs is associated with immunosuppressive effects in HCC [36]. The expression of Fas ligands is induced by the overproduction of VEGFA and prostaglandin E2 (PGE2), resulting in an unnecessary turnover of CD8+ T cells and a decrease in cancer immune responses [35]. Additionally, IDO and IL-2 released by CD14+ DCs contribute to CTL suppression [37].

Furthermore, CTLs play a dual role in the TME of HCC. While they are critical for inducing antitumor immunity, they can also be suppressed by immunosuppressive factors within the TME, contributing to tumor progression. In certain conditions, TME factors suppress CTL activity, limiting their ability to kill cancer cells and sometimes even promoting tumor progression. One major mechanism of CTL suppression involves the Fas-Fas ligand (FasL) pathway. Tumor cells in the TME often overexpress FasL, which binds to Fas receptors on CTLs, triggering their apoptosis. This process is exacerbated by the overproduction of VEGFA and PGE2, which further induce FasL expression on tumor cells. As a result, the excessive turnover of CTLs limits their antitumor activity, weakening the overall immune response [38]. Additionally, immunosuppressive cytokines secreted by both tumor cells and immune cells such as CD14+ DCs further dampen CTL responses. IL-10 suppresses antigen presentation and reduces IFN-γ production by CTLs, effectively blocking their cytotoxic function. IDO, another critical suppressive molecule, depletes tryptophan, an amino acid essential for T cell proliferation and survival. This depletion disrupts CTL activation and limits their ability to exert cytotoxic effects against tumor cells [39]. The TME also imposes metabolic and hypoxic stress that exacerbates CTL suppression. Due to rapid tumor proliferation, HCC cells create a highly competitive environment for nutrients such as glucose. This metabolic competition induces a state of CTL exhaustion, reducing their ability to function effectively. Simultaneously, hypoxia resulting from insufficient blood supply further compromises CTL survival and limits their cytotoxic capabilities [40,41]. These immunosuppressive mechanisms collectively contribute to the failure of CTLs to mount a robust antitumor response in HCC [42]. Understanding how the TME manipulates CTLs is crucial for developing effective immunotherapies aimed at restoring CTL function and enhancing antitumor immunity.

### 3.2. Immunostimulative Roles of DCs

DCs stimulate CD8+ T cells by presenting antigens and are essential for the immune cycle [43]. The liver is rich in various types of DCs, including not only mature cells but also immature and progenitor cells, and immature DCs are often dominant in healthy livers [44]. Liver-resident DCs, originating from the bone marrow, populate regions around central and portal veins, primarily activating the Th1-mediated immune response [45]. The functional characteristics of DCs have recently been elucidated [46,47]. There are two subclasses of DCs: DC1 is the conventional cell population that induces cancer immunity through its ability to stimulate CD8+ T cells and migrate from tumors to LNs. DC2 is involved in antigen presentation on MHC class II molecules and stimulation of CD4 responses, and is present not only in LNs but also in tumors, where it may mediate both immune stimulation and suppression [48]. Furthermore, immunotherapy is ineffective for tumors with an immune desert phenotype, but basic experiments have shown that this may be related to the lack of DCs and absence of T-cell infiltration [31,49].

In HCC, DCs have impaired antigen-presenting functions due to reduce the expression of human leukocyte antigen molecule and defective maturation, resulting in insufficient CD8+ T cell activation and weakened antitumor immunity [50]. Immunosuppressive cytokines in the TME, such as IL-10 and transforming growth factor (TGF)-β, further inhibit DC maturation and promote the differentiation of regulatory T cells (Tregs), which suppress T cell responses and enhance tumor progression. Metabolic stress also plays a critical role in DC dysfunction [51]. Tumor cells outcompete immune cells for essential nutrients, producing high lactate levels that acidify the TME and disrupt DC maturation. Additionally, hypoxic conditions activate hypoxia-inducible factors (HIFs), further suppressing DC function and promoting an immunosuppressive phenotype [51]. The combined effects of these factors significantly limit the ability of DCs to induce effective antitumor immune responses, promoting tumor growth and immune evasion.

### 3.3. Immunosuppression by the Tumor

Cancer cells are also a factor in the TME and suppress the immune system via T cells. For example, PGE2 is a regulator of various immune cells including T cells, and its release activates the cyclooxygenase pathway, which is associated with resistance to immunotherapy [52]. The accumulation of conventional DC1 is suppressed in cancer cells with activated β-catenin signaling [53]. Furthermore, kynurenine, which is produced in excess by cancer cells due to IDO1, tryptophan 2,3-dioxygenase, and adenosine, which are derived from dead cancer cells, are suppressive metabolic products released from cancer cells. Tumors release these metabolites or promote the depletion of amino acids that are essential for T cell function, and the synergistic effects of these metabolic changes influence the TME that suppresses tumor immunity [54]. The release of TGF-β from cancer cells acts suppressively on cancer immunity by promoting the differentiation of Tregs, restricting the proliferation of T stem-like memory cells, and promoting a T cell exclusionary stromal reaction [55,56,57,58]. Notably, cancer cells have an autonomous defense mechanism and protect themselves from damage by T cells by quickly repairing the cell membrane pores created by perforin when T cells release their granules [59].

In the composition of the TME in HCC, not only the tumor itself but also immune cell subsets such as cancer-associated fibroblasts (CAFs), myeloid-derived suppressor cells (MDSCs), tumor-associated macrophages (TAMs), tumor-associated neutrophils (TANs), and Tregs that surround the tumor play important roles in escaping cancer immunity.

### 3.4. Immunosuppressive Roles of CAFs

Fibroblasts support the ECM such as collagen fibers, and in the TME, CAFs play central role in tumor promotion by encouraging the carcinogenesis of normal epithelial cells and giving cancer cell properties similar to those of stromal cells. CAFs originate from activated mesenchymal stem cells in the bone marrow, differentiate from vascular cells in addition to cancer cells, and circulate in the blood.

Three major classes of CAFs are found in human solid tumors: myofibroblastic, inflammatory, and antigen-presenting CAFs [60]. Myofibroblastic CAFs produce ECM components and fibrosis-related molecules, promoting cancer invasion, angiogenesis, and immune evasion by suppressing CD8+ T cells and other immune cells [61,62,63]. Inflammatory CAFs secrete C-C motif chemokine ligand 2 (CCL2), CCL12, and IL-6 and may play an immunosuppressive role [64,65]. Antigen-presenting CAFs express immune regulatory factors and highly express MHC class II molecules, inducing the mobilization of Treg cells [66].

CAFs play a crucial role in the TME, which is useful for suppressing the progression of liver fibrosis and HCC and may become a new therapeutic target [67,68]. As a secondary effect, IDO and PGE2 derived from CAFs attenuate the production of TNF-α and IFN-γ by natural killer cells, which affects the development of HCC [69]. In HCC, bone morphogenetic protein 4 may activate hepatic fibroblasts to secrete cytokines, thereby increasing their infiltration, and is an important regulator of CAF function in HCC [70].

### 3.5. Immunosuppressive Roles of Myeloid Cells

Bone marrow-derived cells proliferate in the TME because of the cytokines and chemokines secreted by cancer cells [71], and MDSCs, macrophages, and monocytes account for approximately half of all cells that comprise the TME [72,73,74]. MDSCs are immature myeloid cells that appear in the blood and local tissues owing to inflammation or cancer and suppress cancer immunity [75]. Hypoxia is a key environmental factor in the TME of various cancer types, including HCC, and MDSCs infiltrate hypoxic regions of HCC tissue via the CCL26/C-X3-C motif chemokine receptor (CXCR) 1 pathway [33]. Another report showed that hypoxia-inducible factor 1-alpha promotes the accumulation of MDSCs by overexpressing ectonucleoside triphosphate diphosphohydrolase 2 in HCC cells [76]. MDSCs may be potential treatment targets for improving immune tolerance in HCC [77].

TAMs infiltrate and accumulate in tumors and are broadly classified into tissue-resident macrophages and monocyte-derived exudative macrophages, which are mobilized by inflammation. The number of these TAMs in the tissue depends on the characteristics of the cancer, and the number of exudative TAMs increases in advanced cancers [78]. When classified from a different perspective according to the activation state of macrophages, TAMs are divided into two subtypes: M1, which undergoes classical activation by TLR ligands and indicators of Th1 responses such as IFN-α, IFN-β, and/or IFN-γ, and M2, which undergoes alternative activation by the Th2 type. The ratio of these two types also differs depending on the characteristics of the cancer: new therapeutic approaches are being developed that selectively deplete tumor-promoting TAMs, directly inhibit their tumor-promoting functions, or reprogram TAM subtypes from tumor promotion to tumor suppression. The role of immunosuppressive TAMs in HCC is attracting attention, and it has been reported that TAMs induce the attraction of Treg cells to tumor tissues and the activation of CTLs by producing various chemokines, such as CCL [79,80], which are expressed in HCC. Furthermore, TAMs have been reported to be associated with the prognosis of patients with HCC; the low expression of CD86+ TAMs and high expression of CD206+ TAMs in HCC tissues are significantly related to clinical stage progression and prognosis [81].

TANs promotes the development of cancer from inflammation and is also strongly involved in the progression and metastasis of the cancer [82]. TANs have opposite effects on tumor biology [83], either anti-tumor or tumor-promoting, and are known to be dependent on the presence of TGF-β [84]. In HCC, the culture supernatant of TANs promotes the migration of Treg and macrophages, mobilizes these cells to HCC tissue, and support tumor progression and sorafenib resistance, indicating the potential of TANs as therapeutic targets [85].

### 3.6. Immunosuppressive Roles of Treg

Tregs are a subset of CD4+ T cells regulated by the transcription factor forkhead box P3. The number of Tregs increases in the peripheral blood and the liver tissue of patients with HCC [86]. Inflammatory signals and long noncoding RNAs (lncRNAs) induce Treg differentiation and promote hepatocarcinogenesis. The overexpression of the epidermal growth factor receptor (lncEGFR) in Tregs activates the downstream activator protein 1/nuclear factor of the activated T cells axis in Tregs, inhibiting their ubiquitination, and consequently promoting immunosuppression in HCC. Furthermore, Tregs may be a potential treatment target for the immunosuppression of HCC, and sorafenib has shown to decrease the number of Treg cells infiltrated into the liver tissue by inhibiting TGF-β signaling [87].

## 4. Role of VEGF in the Cancer Immunity Cycle

VEGF is secreted as a homodimer cross-linked by disulfide bonds, and induces the autophosphorylation of the VEGF receptor (VEGFR), a receptor-type tyrosine kinase. VEGF-A, the most typical member of the VEGF family, binds to both VEGFR-1 and VEGFR-2 expressed on vascular endothelial cells, promoting angiogenesis [88]. In addition to playing a central role in the production of tumor blood vessels, VEGF has recently been recognized its significant impact on the cancer immune cycle. VEGF contributes to immunosuppression by modulating the TME, inhibiting CD8+ T cell infiltration, and promoting immunosuppressive cell populations such as Tregs and MDSCs. The therapeutic strategy of inhibiting VEGF in HCC has evolved not only to target angiogenesis but also to alleviate immunosuppressive mechanisms, thereby synergistically enhancing the effect of ICIs. The effects of VEGF inhibition on the TME are summarized in the scheme (Figure 1). VEGF establishes s an immunosuppressive TME through its influence on antigen priming, CD8+ cell infiltration into tumors, and Treg accumulation.

First, in terms of priming, VEGF inhibits the maturation of DCs and activation of T cells within the LNs, suppresses the function and motility of mature DCs, and enhances PD-L1 expression [89,90]. Thus, it is hypothesized that the inhibition of VEGF promotes priming, accelerates the maturation of immature DCs, enhances the migration and immune function of mature DCs, and activates T cells.

Second, in terms of the tumor infiltration effect of CD8+ T cells, VEGF inhibits the infiltration of CD8+ T cells into tumor tissue by inactivating inflammatory cytokines, reducing intercellular adhesion molecule and vascular cell adhesion molecule, and promoting FasL expression, which induces apoptosis signaling. The inhibition of VEGF is believed to increase these adhesion molecules and reduce FasL expression, promoting tumor tissue infiltration via the desorption, rolling, adhesion, and cloning of CD8+ T cells [90,91]. In a study using 10 pairs of biopsy samples collected before and two weeks after a single dose of a VEGF inhibitor in patients with breast cancer without distant metastasis, bevacizumab promoted the infiltration of mature DCs and CD8+ T cells into tumors [92].

Third, VEGF suppresses cancer immunity by directly or indirectly promoting Treg proliferation. Tregs suppress immune responses centered on T cells through a series of processes in the TME [93,94]. First, CTLA-4 expressed on the surface of Tregs binds to CD80 and CD86 on antigen-presenting cells, resulting in a decrease in the function of antigen-presenting cells and suppression of T-cell activation. Additionally, Tregs inhibit T-cell activation and induce apoptosis by binding to and consuming IL-2, which is essential for T-cell differentiation, proliferation, and survival. Tregs then produce the inhibitory cytokines TGF-β, IL-10, and IL-35, which suppress the functions of T cells and antigen-presenting cells. Furthermore, Tregs produce granzymes and perforin, which lyse and destroy T cells and antigen-presenting cells. Ultimately, Tregs convert adenosine triphosphate to adenosine via CD39 and CD73 and suppress the action of effector T cells and antigen-presenting cells via adenosine receptors. The effect of VEGF inhibition on Tregs has been investigated in various clinical samples. In a study of patients who received bev-combined or non-bev-combined chemotherapy as the primary treatment for colorectal cancer, the proportion of Tregs in the peripheral blood was significantly reduced after treatment in the bev-combined group, indicating that the proportion of Tregs was reduced [95]. In another report on patients with malignant glioma, changes in the number of Tregs were observed in tumors that had not been treated with bev, tumors that had been treated with bev and then resected, and tumors that recurred after bev treatment, indicating that bev restores the immunosuppressive TME [96].

The TME in HCC exhibits profound immunosuppression, driven by the VEGF-mediated recruitment of immunosuppressive cells, limiting antitumor immune responses. However, a potential association with Tregs has been suggested [97]. In patients with HBV and HCV infections and metabolic-related liver disease, HCC prognosis is poorer in cases characterized by a higher abundance of Tregs [98,99,100]. Furthermore, HCC is a hypervascularized tumor with elevated VEGF expression, which correlates with an unfavorable prognosis [101]. The differentiation and proliferation of Tregs induced by VEGF can be broadly divided into direct and indirect effects. VEGF directly promotes Treg proliferation via the VEGF receptor 2 [102]. VEGF indirectly induces Treg differentiation via pathways that inhibit DC maturation and induce MDSC differentiation [103,104]. In HCC and liver metastasis, upregulated VEGF expression suppresses antitumor immunity by promoting Tregs differentiation and proliferation through both direct and indirect pathways, whereas VEGF inhibition attenuate Tregs activation [105]. In HCC treatment, ICI-combined therapies have become the standard treatment strategy, and it is important that CD8+ T cells abundantly infiltrate HCC to achieve maximum treatment effects. Furthermore, the effect of ICI is poor when there are many Tregs suppressing the function of effector T cells such as CD8+ T cells [106]. Therefore, in the TME of HCC, the balance between CD8+ T cells and Tregs is important. To increase the efficacy of ICI, it also makes sense to increase the balance between CD8+ T cells and Tregs by inhibiting VEGF.

While important findings have been accumulated regarding the immunosuppressive effects of VEGF, there remain several issues to be addressed, as follows. The specific molecular mechanisms by which VEGF inhibition affects the interactions between immune cells in HCC are not well understood. Further verification is needed regarding the direct and indirect effects of VEGF inhibition on immunosuppressive cell groups such as Tregs, MDSCs, and TAMs. Furthermore, it has been suggested that the immunomodulatory effects of VEGF inhibition may depend on the molecular subtype of HCC and the patient’s immune profile, but there is limited evidence on this point. The effects of VEGF inhibition in different immune types within the TME, namely, inflammatory, immune exclusion, and immune desert types, may be important in exploring the potential of VEGF inhibition as a predictive biomarker for treatment.

## 5. Promising Therapeutic Strategy for HCC by Inhibiting VEGF

### 5.1. Combination of Anti-VEGF and ICI

In addition to cutting off the blood supply to tumors, VEGF inhibition also enhances antitumor immunity; therefore, combining VEGF inhibitors with ICIs and other immune modulators has the dual effect of promoting sustained antitumor immune responses. The usefulness of combining VEGF-targeting agents with ICIs and the mechanisms that enhance response have been studied in various tumors. In renal cell carcinoma, a combination therapy using bev to inhibit VEGF and atezo to inhibit PD-L1 increased the number of CD8+ T cells in the tumor tissue; increased MHC-1, Th1, and T effector markers; and improved the migration of antigen-specific T cells [107]. In a study investigating the combination of bevacizumab and ipilimumab, anti-CTLA-4 antibodies, in patients with melanoma, extensive morphological changes in CD31+ endothelial cells and extensive infiltration of immune cells were observed in the tumor tissue after a combination therapy [108]. Furthermore, in terms of immune cell infiltration, the combination therapy resulted in an increase in the number of CD8+ T cells and CD163+ macrophages compared to ipilimumab alone.

Like other types of cancer, HCC escapes the cancer-immune system by expressing several immune checkpoint molecules such as PD-1/PD-L1, CTLA-4, T cell immunoglobulin and mucin-containing molecule 3 (TIM-3), and lymphocyte-activation gene 3, and immunotherapy targeting these molecules could be a promising approach [109]. VEGF inhibition also suppresses cancer immunity in HCC, and there is potential for enhancing these immunotherapies. Sorafenib is a multi-targeted kinase inhibitor that has been used for the longest period of basic research, and has been shown to reduce the number of MDSCs in the spleen, bone marrow, and tumors in a liver cancer model [110]. In addition, Sorafenib treatment reduced the density of Tregs in tumors and inhibited their function in a mouse model of liver cancer [111]. Furthermore, anti-VEGF therapy may promote CTL infiltration and activity in other types of cancer [112], and it may also be an important mechanism for influencing the immune cycle in HCC through VEGF inhibition. Basic research on the lenvatinib has shown that the inhibition of VEGF activity reduces TAMs and Tregs in the TME, resulting in a decrease in IL-10 and TGF-β, a decreased expression of T cell exhaustion markers including PD-1 and TIM-3, and an increase in the expression of immunostimulatory cytokines [113,114,115]. These findings form the basis for trials investigating the combination of tyrosine kinase inhibitor (TKI) with anti-PD-1/PD-L1 antibodies.

In HCC, it may be useful to use ICIs in combination with anti-VEGF agents to overcome tumor hypoxia caused by anti-VEGF. In a mouse model of HCC, sorafenib treatment promotes tumor hypoxia and induces stromal cell-derived factor 1α expression and MDSC accumulation in the tumor, but tumor growth is inhibited when CXCR4 receptors are pharmacologically inhibited using the small molecule drug AMD3100 in combination with sorafenib [116]. Furthermore, the expression of PD-L1 increases after sorafenib treatment, and the combination of AMD3100 and the anti-PD-1 antibody significantly delays tumor growth and metastasis by promoting the infiltration of activated CTLs into tumors [117].

### 5.2. Emerging Combination Therapies with ICI and MTA

In addition to the direct inhibition of VEGF, new combinations of ICIs and other MTAs are being researched, and it is hoped that this will lead to overcoming resistance and optimizing therapeutic effects [118]. Table 2 summarizes the results of clinical trials investigating combination therapies of ICIs and MTAs beyond the atezo/bev combination.

The COSMIC-312 Phase III trial evaluated the combination of atezo plus cabozantinib versus sorafenib monotherapy in the patients with advanced HCC. While the combination demonstrated a significant improvement in PFS, interim analyses showed no significant difference in overall survival (OS) between the two groups [119]. A later analysis confirmed that cabozantinib did not improve OS compared to sorafenib [120].

Lenvatinib is also being investigated in multiple trials in combination with ICI. Compared to monotherapy, a recent study reported that combining lenvatinib with anti-PD-1 inhibitors significantly improved PFS and OS compared to anti-PD-1 monotherapy [121]. Other clinical trials confirmed that the combination therapy enhanced the clinical outcomes for patients with advanced HCC [122,123]. In pembrolizumab and lenvatinib combination, a prospective study found that the combination of pembrolizumab and lenvatinib was safe and effective, even in patients with high-risk tumors and compromised liver function [124]. In a large-scale clinical study (LEAP-002 study), the combination of pembrolizumab and lenvatinib showed promising results as a first-line treatment for unresectable HCC, but there was no significant difference in PFS and OS between the groups receiving combination and lenvatinib alone [125]. In another combination, the combination of nivolumab and lenvatinib led to a higher response rate and significant improvements in PFS and OS compared to lenvatinib alone [126]. According to another report, patients with unresectable HCC who received a combination therapy with sintilimab and lenvatinib showed better treatment effects and clinical outcomes than those who received the lenvatinib monotherapy [127]. In post-therapy, lenvatinib as second-line therapy after atezo/bev failure demonstrated clinical efficacy, despite the absence of an immunotherapeutic synergy [128]. Another report has shown that patients who received atezo/bev after lenvatinib treatment may experience rapid tumor growth followed by shrinkage [129].

Regarding the combination of anti-PD-1 antibodies and sorafenib, the combination therapy of nivolumab or pembrolizumab treatment in combination with sorafenib showed a better tumor control rate and prolonged PFS and OS in patients with unresectable HCC compared to anti-PD-1 monotherapy [130]. Furthermore, anti-PD-1 antibodies have also been shown to increase the infiltration of CD4+ and CD8+ T cells and provide a vascular protective effect, which may make subsequent sorafenib treatment more beneficial [131].

**Table 2 ijms-25-13590-t002:** Summary of clinical trials examining the efficacy of the combination therapies of ICI and MTA.

Combinations	Trial	Patient Number	Main Results	Reference
atezolizumab plus cabozantinib	phase III	837	PFS of the combination was better than that of sorafenib, but there was no difference in OS.	[120]
anti-PD-1 plus lenvatinib	retrospective study	94	PFS and OS in the combination therapy were significantly better than those of the anti-PD-1 monotherapy.	[121]
anti-PD-1 plus lenvatinib	retrospective study	213	OS in the combination therapy were significantly better than those of non-combination.	[122]
anti-PD-1 plus lenvatinib	phase II	56	The combination therapy showed good clinical outcomes and safety.	[123]
pembrolizumab plus lenvatinib	prospective study	71	The combination therapy showed good clinical outcomes and safety.	[124]
pembrolizumab plus lenvatinib	phase III	1309	There was no significant difference in PFS or OS between the combination therapy and lenvatinib monotherapy.	[125]
nivolumab plus lenvatinib	retrospective study	87	PFS and OS in the combination therapy were significantly better than those of the lenvatinib monotherapy.	[126]
sintilimab plus lenvatinib	retrospective study	139	The combination therapy showed good clinical outcomes and safety.	[127]
atezolizumab plus bevacizumab plus lenvatinib	retrospective study	101	There were no significant differences in PFS and OS for lenvatinib compared to another MTA.	[128]
anti-PD-1 plus sorafenib	retrospective study	140	The combination therapy showed better tumor control and prolonged prognosis than anti-PD-1 monotherapy.	[130]

ICI, immune checkpoint inhibitor; MTA, molecular target agent; PD-1, programmed cell death 1; PFS, progression free survival; OS, overall survival.

In addition, there are several studies underway on the combination therapy of ICIs with other new MTAs. Anlotinib, a multi-targeted TKI acting on VEGFR, fibroblast growth factor receptor, and platelet-derived growth factor receptor, has emerged as a promising agent in HCC therapy. A recent study has demonstrated its potential to enhance the efficacy of anti-PD-1 immunotherapy. This effect is mediated through suppression of transferrin receptor expression via the VEGFR2/AKT/HIF-1α signaling pathway, accompanied by increased infiltration of CD8+ T cells into the TME, thereby augmenting the antitumor immune response [132,133]. Fruquintinib, a selective VEGFR inhibitor, may enhance the efficacy of ICIs by normalizing tumor vasculature and promoting immune cell infiltration, offering potential as a combination therapy for HCC [134,135]. Rivoceranib, a TKI with the selective inhibition of VEGFR-2, has emerged as a promising agent in combination with camrelizumab for the treatment of HCC. Notably, the findings from the CARES-310 study demonstrated that the rivoceranib and camrelizumab combination therapy significantly prolonged OS compared to sorafenib [136]. Ivonescimab, a bispecific antibody targeting both PD-1 and VEGF, has emerged as a promising therapeutic agent, demonstrating a favorable safety profile and efficacy in the treatment of advanced solid tumors, including HCC [137]. This dual-targeting approach may provide a new strategy for overcoming the challenges of immunosuppressive TME by enabling a more effective blockade of pathways than conventional single antibodies. These combination therapies have the potential to establish a new therapeutic paradigm in the treatment of HCC. Future research and clinical trial advancements are expected to further elucidate their efficacy and safety.

### 5.3. Combination of Anti-VEGF, Immunotherapy, and TACE

In the case of unresectable HCC, therapeutic strategies combining anti-VEGF therapy, immunotherapy, and TACE have been proposed owing to advances in systemic therapy, but the timing of TACE and evidence regarding the suspension and resumption of systemic therapy are not standardized. From the perspective of the effect on TMEs of both blood vessels and cancer immunity when used in combination with TACE, it seems that there is significance in combining atezo/bev therapy. Basic research has shown that performing TACE in HCC causes changes in the TME owing to the release of VEGF and cancer antigens caused by tumor hypoxia [138], which leads to an immunosuppressive TME owing to an increase in Tregs and MDSC [108,139]. Another report showed that after TACE, cancer antigens are released, the number of CD8+ T cells increases, and CTLs infiltrate the tumor, which may affect the therapeutic effects of atezo. To overcome these effects [140], it is important to use it in combination with drugs that inhibit VEGF, such as bev.

The standard treatments for unresectable HCC are TACE for BCLC stage B and systemic therapy for stage C [141]. However, in cases where the effect of TACE cannot be expected owing to the Up-to-7 criteria or tumor morphology or in cases where liver function is likely to deteriorate [142], it is useful to implement systemic therapy and, in some cases, add TACE to the treatment strategy. It has also been reported that the effect of TACE can be enhanced by anti-VEGF inhibitors by first administering atezo/bev therapy [143,144]. Furthermore, the number of patients with BCLC Stage C in whom TACE is combined with systemic therapy is increasing. Controlling intrahepatic lesions is an important prognostic factor, even in cases with extrahepatic metastases [145], and regardless of the clinical stage, there is significance in combining anti-VEGF, immunotherapy, and TACE.

Furthermore, the efficacy of combining ICIs and MTAs has also been reported in hepatic arterial infusion chemotherapy (HAIC), which is often performed for more advanced HCC where the effect of single TACE is thought to be limited. The combination of pembrolizumab, lenvatinib, and HAIC resulted in a median PFS of 10.9 months and a median OS of 17.7 months in patients with histologically PD-L1-positive HCC [146]. According to a meta-analysis, the combination of HAIC and anti-PD-1/anti-PD-L1 was associated with an increased incidence and severity of adverse events, but also improved treatment outcomes and prognosis for patients with unresectable HCC compared to the combination of anti-angiogenic agents and anti-PD-1/anti-PD-L1 [147].

As shown in the above studies, a combination therapy using VEGF inhibitors and ICIs is a promising treatment strategy for HCC, but there are still some important issues to be addressed. The results of combination therapy in clinical trials are not always consistent, and it is necessary to identify predictors of treatment efficacy. For example, there are studies that show an extension of PFS but do not observe a significant improvement in OS. This suggests the need for personalized medicine based on the patient’s immune profile and the molecular characteristics of HCC. Furthermore, treatment in ICIs is expensive, and it is unclear how treatment efficacy and cost may differ among ICIs with similar mechanisms of action in HCC. Future studies comparing cost-effectiveness are needed.

## 6. Conclusions

The combination of ICIs and VEGF inhibition has emerged as a promising therapeutic strategy for HCC. This review underscores the immunomodulatory effects of VEGF inhibition, including vascular normalization, immune cell recruitment, and the suppression of immunosuppressive cells such as Tregs and MDSCs. Despite these advances, key challenges remain. First, the identification of robust molecular biomarkers is essential for predicting treatment responses and guiding personalized therapy. Second, optimizing the interplay between VEGF inhibitors and ICIs necessitates a deeper understanding of immune cell dynamics and TME remodeling. Third, comprehensive molecular profiling of resistant tumors is imperative to uncover novel therapeutic targets and refine treatment protocols. Addressing these challenges will enhance the efficacy of the combination therapies in HCC. Further research into the molecular mechanisms governing vascular and immune modulation could drive the development of more effective, individualized treatment strategies.

## Figures and Tables

**Figure 1 ijms-25-13590-f001:**
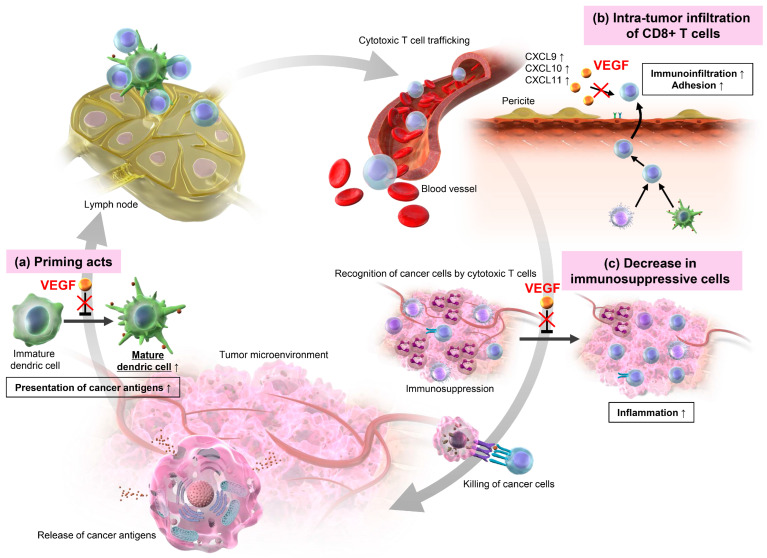
Effects of vascular endothelial growth factor (VEGF) inhibition in the cancer immunity cycle. (**a**) VEGF inhibition promotes priming, enhances the maturation of immature dendritic cells, increases the migration ability and immune function of mature dendritic cells, and activates T cells. (**b**) VEGF inhibition increases adhesion molecules including C-X-C motif chemokine ligand (CXCL) and promotes infiltration into tumor tissue through the detachment, rolling, adhesion, and cloning of CD8+ T cells. (**c**) Inhibiting VEGF reduces immunosuppressive cells such as regulatory T cell and myeloid-derived suppressor cells and lifts the immunosuppressive state.

**Table 1 ijms-25-13590-t001:** Summary of the tumor immunity cycle.

Cancer Immunity Cycle	Key Components	Main Phenomenon
Step 1	Dead cancer cells	Necrosis or release of cancer antigens due to treatment effects
Step 2	DCs	Cancer antigen uptake and migration to LNs
Step 3	CTLs	Promoting priming and activation
Step 4	CTLs	Circulation within blood vessels and migration to tumors
Step 5	CTLs	Infiltration into tumors
Step 6	CTLs	Recognition of cancer cells via TCRs
Step 7	CTLs	Attacking cancer cells

DCs, dendric cells; CTLs, cytotoxic T lymphocyte; LNs, lymph nodes; TCRs, T cell receptors.

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
