# Peer review of "Immune Microenvironment and the Effect of Vascular Endothelial Growth Factor Inhibition in Hepatocellular Carcinoma"

_ijms, 2024, doi:10.3390/ijms252413590_

Round 1
Reviewer 1 Report
Comments and Suggestions for Authors
This article reviews the role of vascular endothelial growth factor (VEGF) in the tumor immune microenvironment of hepatocellular carcinoma (HCC). It highlights VEGF's immunosuppressive effects, including limiting CD8+ T-cell infiltration and promoting regulatory T-cell proliferation. The paper emphasizes the therapeutic potential of combining VEGF inhibitors with immune checkpoint inhibitors to enhance anti-tumor immunity. Lastly, it discusses current clinical trials and strategies integrating VEGF inhibition with systemic therapies for improved outcomes in HCC treatment.
Major revisions:
- The paper offers a detailed overview of the role of VEGF in the tumor immune microenvironment of hepatocellular carcinoma (HCC). However, the novelty of the findings is limited as the discussion largely reiterates previously established concepts without significant new insights or original data. The authors must provide novel experimental data or unique perspectives that advance the current understanding of VEGF inhibition's role in HCC. Additionally, the review neglects emerging therapies, including Anlotinib, Fruquintinib, Ivonescimab, and Rivoceranib, which represent critical developments in the field and should be thoroughly addressed to provide a more complete and relevant analysis.
- The manuscript extensively references prior research but fails to integrate these findings into a cohesive framework that advances the field. Key studies are discussed in isolation without critical synthesis. Incorporate a more critical analysis of existing studies, identifying gaps in the literature and clearly explaining how this paper addresses those gaps.
- The discussion of VEGF inhibition in the tumor microenvironment lacks depth, particularly in exploring the complex interplay between other molecular pathways and VEGF's role in HCC progression. Key emerging therapies and experimental findings that could enrich the narrative are either missing or underexplored.
- The manuscript's structure is overly complex, with redundant sections that obscure the main arguments. Streamline sections by reducing redundancy and improving transitions between the tumor microenvironment, cancer-immunity cycle, and VEGF inhibition topics.
- The manuscript does not convincingly translate its findings into actionable insights for clinical practice. While therapeutic strategies are mentioned, there is no detailed exploration of their practical challenges, risks, or limitations. The role of VEGF inhibition in combination therapies is discussed, but the manuscript does not address the significant barriers to implementation, such as cost, patient stratification, or resistance mechanisms.
Minor revisions:
- The figures provided, such as the role of VEGF in the cancer immunity cycle, lack sufficient detail and originality. These diagrams do not add substantial value to the understanding of the concepts discussed. Redesign figures to include unique visual insights or data that enhance the reader's comprehension.
- The manuscript contains grammatical errors and poorly constructed sentences that hinder readability (example: "VEGF inhibition is thought that VEGF inhibition suppress Tregs"). Proofread and revise the text for grammatical accuracy and readability.
- Line 16 and line 53: Tremelimumab + durvalumab, not durvalumab + durvalumab
Comments on the Quality of English LanguageThe manuscript contains grammatical errors and poorly constructed sentences that hinder readability (example: "VEGF inhibition is thought that VEGF inhibition suppress Tregs"). Proofread and revise the text for grammatical accuracy and readability.
Author Response
Response to reviewer 1 comments
- The paper offers a detailed overview of the role of VEGF in the tumor immune microenvironment of hepatocellular carcinoma (HCC). However, the novelty of the findings is limited as the discussion largely reiterates previously established concepts without significant new insights or original data. The authors must provide novel experimental data or unique perspectives that advance the current understanding of VEGF inhibition's role in HCC. Additionally, the review neglects emerging therapies, including Anlotinib, Fruquintinib, Ivonescimab, and Rivoceranib, which represent critical developments in the field and should be thoroughly addressed to provide a more complete and relevant analysis.
Response: Thank you very much for your comments.
This is a duplicate of my response to your second comment below, but I have added some additional perspectives that may deepen the current understanding of the role of VEGF inhibition in HCC.
To enhance the positioning and issues of previous studies, we revised the introduction to clarify research gaps and objectives. We highlighted the limited understanding of VEGF in tumor immunosuppression, the combined effects with ICIs and VEGF inhibition. Summaries of existing findings, limitations, and implications were added at the end of sections 4 and 5. We also outlined future challenges, including cell signaling analysis within biomarker discovery for personalized therapy.
In conclusion, combining ICIs with VEGF inhibition shows promise for HCC treatment by enhancing vascular normalization, immune cell recruitment, and suppressing immunosuppressive cells. However, challenges remain, such as identifying predictive biomarkers, optimizing VEGF-ICI interactions, and profiling resistant tumors to refine treatments. Addressing these will advance personalized HCC therapies.
Furthermore, as you pointed out, Anlotinib, Fruquintinib, Ivonescimab, and Rivoceranib are attracting attention as combination therapies with ICIs in the treatment of HCC. Anlotinib, a multi-targeted tyrosine kinase inhibitor acting on VEGFR, FGFR, and PDGFR, has emerged as a promising agent in HCC therapy. A recent study has demonstrated its potential to enhance the efficacy of anti-PD-1 immunotherapy. This effect is mediated through suppression of transferrin receptor expression via the VEGFR2/AKT/HIF-1α signaling pathway, accompanied by increased infiltration of CD8+ T cells into the TME, thereby augmenting the antitumor immune response (Song F, et al. Clin Transl Med 14(8): e1738, 2024). Fruquintinib, a selective VEGFR tyrosine kinase inhibitor (TKI), may enhance the efficacy of ICIs by normalizing tumor vasculature and promoting immune cell infiltration, offering potential as a combination therapy for HCC (Shao G, et al. Journal of Clinical Oncology 42(3): 499, 2024). Rivoceranib, a TKI with selective inhibition of VEGFR-2, has emerged as a promising agent in combination with camrelizumab for the treatment of HCC. Notably, findings from the CARES-310 study demonstrated that the rivoceranib and camrelizumab combination therapy significantly prolonged OS compared to sorafenib (Qins S, et al. Lancet 402: 1133-1146, 2023). Ivonescimab, a bispecific antibody targeting both PD-1 and VEGF, has emerged as a promising therapeutic agent, demonstrating a favorable safety profile and efficacy in the treatment of advanced solid tumors, including HCC (Frentzas S, et al. J Immunother Cancer 12(4): 2024). This dual-targeting approach may provide a new strategy for overcoming the challenges of immunosuppressive TME by enabling more effective blockade of pathways than conventional single antibodies. These combination therapies have the potential to establish a new therapeutic paradigm in the treatment of HCC. Future research and clinical trial advancements are expected to further elucidate their efficacy and safety.
We have revised the relevant parts of the article to emphasize these points. Revisions in the manuscript are highlighted in yellow.
- The manuscript extensively references prior research but fails to integrate these findings into a cohesive framework that advances the field. Key studies are discussed in isolation without critical synthesis. Incorporate a more critical analysis of existing studies, identifying gaps in the literature and clearly explaining how this paper addresses those gaps.
Response: To strengthen the critical synthesis and positioning of the research, we first strengthened the introduction to clarify the research gaps and objectives. Specifically, we added information on the lack of a comprehensive understanding of the mechanism of immunosuppression release by VEGF inhibition, the uncertainty regarding the mechanism of action of the combined effects of VEGF inhibition and ICIs, and the systematic organization of the characteristics of the TME of HCC and treatment targets. Next, we added a paragraph at the end of each chapter on 4. The role of VEGF and 5. Promising Therapeutic Strategy for HCC that briefly summarizes the results of existing research, limitations, and implications for new treatment strategies. Finally, we listed unresolved issues and topics that require additional verification as future challenges. Specifically, we added information on signal transduction analysis between cells within the tumor regarding immune-modifying effects and the search for biomarkers for personalized treatment.
In conclusion, the combination of ICIs and VEGF inhibition has emerged as a promising therapeutic strategy for HCC. This review highlighted the immunomodulatory effects of VEGF inhibition, including vascular normalization, immune cell recruitment, and suppression of immunosuppressive cells such as Tregs and MDSCs. Despite these advances, key challenges remain. First, identifying robust molecular biomarkers is essential for predicting treatment responses and guiding personalized therapy. Second, optimizing the interaction between VEGF inhibitors and ICIs requires a deeper understanding of immune cell dynamics and tumor microenvironment remodeling. Third, comprehensive molecular profiling of resistant tumors is needed to uncover new therapeutic targets and refine treatment protocols. Addressing these issues will enhance the efficacy of the combination therapies in HCC. Further research into the molecular mechanisms governing vascular and immune modulation could drive the development of more effective, individualized treatment strategies.
The relevant part of the article has been revised.
- The discussion of VEGF inhibition in the tumor microenvironment lacks depth, particularly in exploring the complex interplay between other molecular pathways and VEGF's role in HCC progression. Key emerging therapies and experimental findings that could enrich the narrative are either missing or underexplored.
Response: We have added a description of the main treatment methods and new treatment strategies for VEGF inhibition to Section 5. The details are the same as the responses to the comments above.
- The manuscript's structure is overly complex, with redundant sections that obscure the main arguments. Streamline sections by reducing redundancy and improving transitions between the tumor microenvironment, cancer-immunity cycle, and VEGF inhibition topics.
Response: We have revised the redundancy of Sections 2 to 4 so that the structure of the text is concise and logical, and the roles of immune cells, the cancer immunity cycle, and the interrelationship of VEGF inhibition are clearly connected. Revisions in the manuscript are highlighted in yellow.
- The manuscript does not convincingly translate its findings into actionable insights for clinical practice. While therapeutic strategies are mentioned, there is no detailed exploration of their practical challenges, risks, or limitations. The role of VEGF inhibition in combination therapies is discussed, but the manuscript does not address the significant barriers to implementation, such as cost, patient stratification, or resistance mechanisms.
Response: As noted above, we have added practical challenges and limitations related to patient’s stratification and treatment resistance in section 5 and the conclusion. Furthermore, as you point out, treatment in ICIs is expensive, and drug prices vary widely for similar mechanisms. The cost-effectiveness of ICI combination therapy in HCC remains unclear, and future studies comparing cost-effectiveness are needed.
In limitations, the relevant part of the article has been revised.
- The figures provided, such as the role of VEGF in the cancer immunity cycle, lack sufficient detail and originality. These diagrams do not add substantial value to the understanding of the concepts discussed. Redesign figures to include unique visual insights or data that enhance the reader's comprehension.
Response: Figure 1 was newly created to include not only an overview of the immune cycle, but also the effects of VEGF at each checkpoint within it. Furthermore, this figure was redesigned and modified to emphasize the role of VEGF inhibition in the cancer immune cycle. If there are any further improvements that need to be made, please let us know.
- The manuscript contains grammatical errors and poorly constructed sentences that hinder readability (example: "VEGF inhibition is thought that VEGF inhibition suppress Tregs"). Proofread and revise the text for grammatical accuracy and readability.
Response: We corrected these errors and carefully checked again for other errors. We also asked the English proofreading company to revise the text again to make it grammatically correct. Revisions in the manuscript are highlighted in yellow.
- Line 16 and line 53: Tremelimumab + durvalumab, not durvalumab + durvalumab
Response: We corrected these errors and carefully checked again for other errors. We also asked the English proofreading company to revise the text again to make it grammatically correct. Thank you very much for your careful peer review.

Reviewer 2 Report
Comments and Suggestions for Authors
Dear Authors,
The purpose of this review was to “outline the current knowledge of the immune microenvironment of the immune microenvironment in HCC and summarize the role of VEGF in the immune cycle of the tumor". The paper is very similar to your previous publication from 2021 with similar Figure [ref. no. 21], which addressed most of the issues presented in the current review. The innovation of the current paper should be emphasized.
My other comments:
I recommend expanding the Introduction subsection, as well as the data on the role of VEGF in liver TME in HCC. The Abstract and the Introduction to the paper lack a reference to the title of the paper, i.e. marking the role of the immune microenvironment of HCC in the context of the topics discussed further. The Introduction to the paper should be expanded to include the role of VEGF in TME in HCC (with additional figure), and only later the thread on anti-VEGF treatment. In Chapter 2 - Table 1 should be cited, not Figure 1 (line 74). Please remove the cited own paper 21, as it does not add anything new to the current work, or alternatively state what the authors previously described. In subsection 3 - I understand that the role of CTLs in TME is dual, not just immunostimulating (line 111 vs. 126). In subsection lines 136-141, reference to the third subtype of DCs is missing (line 136).
In subsection 5, lines 331-337 have repetitive content, please delete.
Other errors noted:
in the abstract (line 16) and line 53 “durvalumab” is repeated twice, please add another drug (tremelibumab?); line 34 - the word “virus” in lower case; line 78 - should be “complex”; line 321 - drop the letter “t”,; line 325 - add CD8+ “T cells”; in subheading 5.2 I would give the whole words MTA and ICI anie their abbreviations. ; line 361 - should be “analysis”; line 367 - “reserch” in lower case; I recommend shortening the descriptions of treatment effects, write what was shown right away, not the same thing twice - compare line 374 - please remove “investigating the efficacy of adding...”; Under Table 2 - please add an explanation of the abbreviation ICI, or write it in the header; line 409 - should be “for BCLC stage B”; line 423 - should be “HCC” not nHCC.
Table of References - generally needs correction, especially in the number of names of papers cited, there are too many. Please check with the editors' recommendations.
The paper can be accepted for publication after necessary additions and consideration of text corrections.
Author Response
Response to reviewer 2 comments
- The purpose of this review was to “outline the current knowledge of the immune microenvironment of the immune microenvironment in HCC and summarize the role of VEGF in the immune cycle of the tumor". The paper is very similar to your previous publication from 2021 with similar Figure [ref. no. 21], which addressed most of the issues presented in the current review. The innovation of the current paper should be emphasized.
Response: Thank you very much for your comments.
As combination therapies such as atezolizumab/Bevacizumab combination therapy, which uses ICIs and VEGF inhibitors, become more widely used, the significance of VEGF inhibition is being recognized not only in terms of its effect on suppressing angiogenesis, but also in terms of its effect on the cancer immune cycle. The tumor microenvironment (TME) of HCC is a complex and dynamic system composed of tumor cells, immune cells, endothelial cells, fibroblasts, stromal cells, and extracellular matrix (ECM) components. In this regard, VEGF plays an important role not only in promoting angiogenesis, but also in regulating immune responses, promoting tumor progression, and causing treatment resistance. Therefore, elucidating the complex mechanisms of the immune microenvironment of HCC, rather than simply inhibiting angiogenesis by inhibiting VEGF, may lead to the development of therapeutic targets and the overcoming of treatment resistance.
In the past, there have been reports of research and review articles related to the immune microenvironment in tumors. However, the novelty of this research is that it examines the potential significance of dual inhibition of angiogenesis by VEGF and immune-related molecules, and summarizes the latest treatment strategies to enhance therapeutic efficacy.
Figure 1 is not just a summary of the immune cycle, but also includes the effects of VEGF inhibition at each checkpoint within it. The figure has been newly created to emphasize the role of VEGF inhabitation in the cancer immune cycle. If there are any further improvements you would like to see, please let us know.
As you pointed out, we have corrected the relevant parts of the article to emphasize these points. Revisions in the manuscript are highlighted in yellow.
- I recommend expanding the Introduction subsection, as well as the data on the role of VEGF in liver TME in HCC. The Abstract and the Introduction to the paper lack a reference to the title of the paper, i.e. marking the role of the immune microenvironment of HCC in the context of the topics discussed further. The Introduction to the paper should be expanded to include the role of VEGF in TME in HCC, and only later the thread on anti-VEGF treatment. In Chapter 2 - Table 1 should be cited, not Figure 1 (line 74). Please remove the cited own paper 21, as it does not add anything new to the current work, or alternatively state what the authors previously described. In subsection 3 - I understand that the role of CTLs in TME is dual, not just immunostimulating (line 111 vs. 126). In subsection lines 136-141, reference to the third subtype of DCs is missing (line 136).
Response: As you pointed out, the TME of HCC is a complex and dynamic system composed of tumor cells, immune cells, endothelial cells, fibroblasts, stromal cells, and ECM components. In this context, VEGF plays a pivotal role not only in promoting angiogenesis but also in modulating immune responses, fostering tumor progression, and inducing therapeutic resistance. Firstly, VEGF promotes an-giogenesis and changes the tissue environment. In HCC, the new blood vessels formed by VEGF are often structurally abnormal and highly leaky, which increases the interstitial pressure within the tumor and impairs drug delivery and creates a hypoxic environment (Andrew XZ, et al. Nat Rev Clin Oncol 8(5): 292-301, 2011). This hypoxic state further stimulates the production of VEGF, forming a feedback loop that exacerbates angiogenesis and tumor growth. Secondly, VEGF promotes tumor immune escape by mobilizing and suppressing immune cells and reducing the function of antigen-presenting cells, thereby forming an immuno-suppressive TME (Yuchen Q, et al. J Cell Mol Med 27(4): 538-552, 2023). Thirdly, VEGF modulates communication between tumor cells and immune cells. Activation of the VEGF signaling pathway causes abnormal secretion of cytokines and chemokines, and regulates the mobilization and localiza-tion of immune cells (Xiaoting L, et al. MedComm 5(2): e474, 2020). In addition, VEGF induces fibroblast activation and ECM remodeling, accelerating tumor cell invasion and metastasis (Arya MR, et al. Cell Rep Med 4(9): 101170, 2023). Therefore, elucidating the complex mechanisms of the immune microenvironment of HCC, rather than simply inhibiting angiogenesis through VEGF inhibition, may lead to the develop-ment of therapeutic targets and the overcoming of treatment resistance. The above description has been added to the introduction.
In addition, the citation in Chapter 2 regarding the immune cycle has been corrected in Table 1. As you pointed out, my paper 21 does not add any new information to this research, so I have deleted it from the article.
TME in HCC suppresses CTLs, limiting their anti-tumor function and promoting tumor progression. The main mechanisms include apoptosis of CTLs via the Fas-FasL pathway, immunosuppressive cytokines such as IL-10 and IDO, and metabolic and hypoxic stress that exhausts CTLs. Understanding and targeting these inhibitory mechanisms can enhance the efficacy of immunotherapy. The details of these have been added to the relevant section of 3.1. Immunostimulative Roles of CTLs.In addition, in the section 3.2. Immunostimulative Roles of DCs, the main two subtypes of DCs have been corrected in the main text. We have also added a description of the anti-tumor immune response of DCs in HCC.
- In subsection 5, lines 331-337 have repetitive content, please delete.
Response: We corrected these errors and carefully checked again for other errors. We also asked the English proofreading company to revise the text again to make it grammatically correct.
- In the abstract (line 16) and line 53 “durvalumab” is repeated twice, please add another drug (tremelibumab?); line 34 - the word “virus” in lower case; line 78 - should be “complex”; line 321 - drop the letter “t”,; line 325 - add CD8+ “T cells”; in subheading 5.2 I would give the whole words MTA and ICI anie their abbreviations. ; line 361 - should be “analysis”; line 367 - “reserch” in lower case; I recommend shortening the descriptions of treatment effects, write what was shown right away, not the same thing twice - compare line 374 - please remove “investigating the efficacy of adding...”; Under Table 2 - please add an explanation of the abbreviation ICI, or write it in the header; line 409 - should be “for BCLC stage B”; line 423 - should be “HCC” not nHCC.
Response: We corrected these errors and carefully checked again for other errors. We also asked the English proofreading company to revise the text again to make it grammatically correct.
Furthermore, to make it easier for readers to understand the descriptions of the treatment effects in Section 5.2., the parts that were superfluous were revised and simplified.
- Table of References - generally needs correction, especially in the number of names of papers cited, there are too many. Please check with the editors' recommendations.
Response: After carefully checking the journal's guidelines, we confirmed that the citation style is in line with the journal's formatting recommendations. Also, based on past articles, the number of references could be 100-200 for a review article of this volume.
However, to ensure clarity and relevance, we re-examined the references and prioritized those that most directly support the main arguments of the paper. Please let us know if further adjustments are needed.
Thank you very much for your careful peer review.

Round 2
Reviewer 1 Report
Comments and Suggestions for Authors
The majority of the requested modifications have been implemented; however:
- redundancies in the tumor immune cycle discussion still persist in the samples.
- grammatical adjustments are observed, though some awkward phrasings persist.
Author Response
Response to reviewer 1 comments
The majority of the requested modifications have been implemented; however:
- redundancies in the tumor immune cycle discussion still persist in the samples.
- grammatical adjustments are observed, though some awkward phrasings persist.
Response:
We sincerely appreciate the reviewer’s thoughtful feedback on our manuscript. In response to the comments provided, we have carefully revised the relevant sections to improve clarity and readability. Specifically, we streamlined the discussion of the tumor immune cycle by removing redundant content and consolidating key points to enhance logical flow. Additionally, we conducted a comprehensive language review, focusing on correcting grammatical errors and refining awkward phrasings to ensure smoother readability. Revisions in the manuscript are highlighted in yellow.
We trust that these revisions adequately address the reviewer’s concerns and contribute to the overall quality of the manuscript. Thank you again for your valuable suggestions, which have been instrumental in strengthening our work.

Reviewer 2 Report
Comments and Suggestions for Authors
You responded to my all doubts and questions posed by myself
following reading of the first version of the paper.
The other data or corrections have been introduced everywhere where
they were required. In this form the paper can be accepted for publication.
Please correct only the sentence in lines 412-413 - it should be written on one line and subsection 5.2 - the title of the main words in capital letters as everywhere in the work.
Author Response
Response to reviewer 2 comments
You responded to my all doubts and questions posed by myself following reading of the first version of the paper. The other data or corrections have been introduced everywhere where they were required. In this form the paper can be accepted for publication. Please correct only the sentence in lines 412-413 - it should be written on one line and subsection 5.2 - the title of the main words in capital letters as everywhere in the work.
Response:
Thank you very much for your careful peer review.
As you pointed out, we have revised the relevant section. We also asked the English proofreading company to revise the text again to make it grammatically correct. Revisions in the manuscript are highlighted in yellow.
We trust that these revisions adequately address the reviewer’s concerns and contribute to the overall quality of the manuscript. Thank you again for your valuable suggestions, which have been instrumental in strengthening our work.
